# Behavioral Theory-Based Framework for Prediabetes Self-Care System—Design Perspectives and Validation Results

**DOI:** 10.3390/ijerph18179160

**Published:** 2021-08-31

**Authors:** Suthashini Subramaniam, Jaspaljeet Singh Dhillon, Wan Fatimah Wan Ahmad

**Affiliations:** 1Faculty of Information and Communication Technology, Universiti Tunku Abdul Rahman (UTAR), Kampar 31900, Malaysia; suthashini@utar.edu.my; 2College of Computing & Informatics, Universiti Tenaga Nasional (UNITEN), Kajang 43000, Malaysia; 3Computer & Information Sciences Department, Universiti Teknologi Petronas (UTP), Seri Iskandar 32610, Malaysia; fatimhd@utp.edu.my

**Keywords:** behavior change techniques, behavioral change theories, diabetes prevention, health informatics, prediabetes, self-care, self-empowerment, stages of change

## Abstract

Type 2 diabetes (T2D) is a chronic condition that can lead to many life-threatening diseases. Prediabetes is defined as a state in which blood glucose levels are elevated but not high enough to be diagnosed as diabetes. This stage can be reversible with appropriate lifestyle and dietary modifications. Existing solutions are mostly developed to deal with T2D instead of preventing it in the first place. In this study, we propose a framework to aid in the development of self-care systems to prevent T2D, which integrates behavioral change theories and techniques and offers features, such as goal setting, activity planning, and health monitoring. We then assessed the feasibility of a prediabetes self-care system designed based on the proposed framework. Quantitative and qualitative methods were adopted in evaluating i-PreventDiabetes, a prototype. Numerous aspects of the prototype were evaluated, including (1) its effectiveness in assisting individuals with prediabetes in improving their health management behaviors, (2) its effect on users’ attitudes toward diabetes prevention, (3) users’ motivation levels, (4) user acceptability of the system, and (5) user experience. Users viewed i-PreventDiabetes positively and experienced a positive change in their attitude toward their health. Diabetes prevention systems, such as i-PreventDiabetes, have the potential to increase self-care behaviors among individuals with prediabetes, enabling them to manage their lifestyle and nutrition more effectively to avert a variety of potentially fatal conditions.

## 1. Introduction

Diabetes is defined as a non-communicable illness that affects a large number of people. Type 2 diabetes (T2D) has been determined to be caused by the body’s inability to use insulin effectively, and it affects the vast majority of individuals with diabetes [1]. Preventing T2D is a vital step in preventing chronic illnesses. T2D may be avoided during the prediabetes stage by modifying one’s diet and lifestyle. In recent years, the number of people diagnosed with T2D has risen dramatically. Prediabetes is a stage of diabetes in which the blood glucose level is elevated above normal but not high enough to be diagnosed as diabetes. T2D may be avoided or delayed at this time by observing a healthy diet and increased physical activity [2,3]. Human behavior is critical in establishing a healthy diet and lifestyle. When positive behavior is ingrained, a new, healthier habit for a better life then develops. Diet, increased physical activity, and behavior therapy are the three critical components of lifestyle change [4].

Research and development efforts are mostly focused on the design and development of diabetes management systems rather than diabetes prevention [5,6,7]. Diabetes management systems are intended for those who have already been diagnosed with diabetes, whereas diabetes prevention systems are designed for people who have not yet been diagnosed with diabetes but are at the risk of developing it. Existing diabetes self-care systems encounter challenges in convincing individuals with prediabetes to make beneficial lifestyle and dietary changes [8,9,10]. These systems were not developed by healthcare professionals or academics [11,12] and do not incorporate behavioral change theories, resulting in poor diabetes control adherence and unsustainable intervention use [8,13].

Self-care has been deemed crucial for disease prevention. Health consumers need to be empowered to take care of their health and make informed lifestyle modification decisions [14,15]. Self-care behavior is critical for achieving favorable health outcomes [16]. A combination of technology and self-care can assist people with prediabetes to manage high blood glucose levels. Behavior change techniques (BCTs) can aid in the enhancement of behavior change interventions [17,18]. Goal setting, action planning, self-monitoring, problem-solving, and goal review are examples of BCTs that can be incorporated into interventions to reduce one’s blood glucose levels. When BCTs are carefully integrated in interventions, substantial changes in health behaviors and psychological outcomes are expected [18].

A system becomes less effective if there is minimal commitment from the target users [19]. Eliciting user needs from targeted users is critical for designing successful healthcare systems to encourage continued use of any healthcare intervention and offer high-quality treatment to consumers [20]. Given that many people with prediabetes do not take their conditions seriously or lack the self-motivation or knowledge to make the necessary behavioral changes [21], it is thus critical to gather healthcare professionals’ perspectives on the essential functions to include in a T2D prevention system.

There is a need for systems that integrate self-care, health behavioral change, and user needs to assist health consumers in managing their own health to avoid the onset of diabetes, rather than treating the condition, as most systems do. In this paper, we propose a design framework for self-care systems developed for prediabetes. We use the framework to design a system and then assess the system’s feasibility. We explain in detail the design of a prototype, i-PreventDiabetes, which integrates health behavioral change theories, BCTs, and user requirements based on this framework [22]. The system was assessed in order to draw conclusions about its feasibility and acceptance by individuals with prediabetes. Numerous aspects of the system were evaluated, including (1) its effectiveness in assisting people with prediabetes in improving their health management behaviors, (2) its effect on users’ attitudes toward diabetes prevention, (3) users’ motivation levels, (4) user acceptability of the system, and (5) user experience. We try answering the research questions “how best to design a self-care system to prevent T2D that integrates behavioral change theories to promote diabetes prevention, covers essential features to assist in reducing blood glucose levels, and is well accepted by people with prediabetes?”

## 2. Design

i-PreventDiabetes was designed bottom-up, through the eyes of people with prediabetes, with the aim of enabling them to be more proactive in managing their own blood glucose levels. We have employed a patient-centered approach in developing i-PreventDiabetes by working closely with individuals with prediabetes and healthcare professionals from the outset. In this section, we describe the proposed framework design and i-PreventDiabetes system prototype. The framework provides a complete view of the suggested solution to overcome the shortcomings of existing systems that are primarily designed for patients with T2D to manage their conditions and not as diabetes prevention interventions. The system was developed to assess the overall concept, content, and feasibility with people who are prediabetic.

### 2.1. Framework Design

The framework was developed in accordance with behavioral change theories and user needs. Numerous focus group discussions (FGDs) were conducted to ascertain the needs of people with prediabetes in terms of self-care and T2D prevention [21]. Additionally, we conducted semi-structured interviews with healthcare experts to elicit their knowledge and perspectives on self-care system design [23]. In light of these considerations, the framework was developed [22] and served as the foundation for developing a system prototype to validate the framework’s core features.

The suggested framework’s general structure is shown in Figure 1, and consists of many components with the overarching goal of enabling people with prediabetes to take control of their own blood glucose levels. Prochaska and DiClemente’s transtheoretical model (TTM), commonly known as the stages of change model, is used as the primary behavioral change theory in this framework [24,25]. This theory demonstrates the phases of behavior modification in a person with prediabetes and focuses on the individual’s decision-making abilities [26]. Pre-contemplation, contemplation, preparation, action, and maintenance are the phases of behavior change [24]. The stages are detailed in Table 1. Users might regress to any stage until they achieve a stable lifestyle. Additionally, the health belief model (HBM), theory of planned behavior (TPB), and attitude formation models have been integrated [27].

The first part of the framework discusses the components that contribute to a specific behavior: triggers, a positive attitude, education and awareness, motivation, and commitment. Table 2 presents the contributing factors. Additionally, the framework incorporates BCTs and self-care behaviors associated with the American Association of Diabetes Educators (AADE7) [17,18,28]. The BCTs are included as features in the prototype (listed in Table 3). These BCTs/features were identified via discussions with individuals with prediabetes and diabetic patients, as well as healthcare experts [21,23].

### 2.2. Prototype Design

Based on the framework presented in Figure 1, we have designed a prototype in order to demonstrate the utility of the key features and provide evaluation results with the help of user studies. Most of the BCTs from the framework are included as features presented in the system. i-PreventDiabetes has been developed by considering the self-care abilities in mind. Upon completion of its initial version, a formative evaluation was performed with 20 participants to identify the usability barriers and to assess the user experience [29]. Constructive feedback obtained was used to enhance the system.

Figure 2 illustrates the dashboard of the system, which shows the features (described in Table 3) incorporated into the system. The system also encompasses a reward system consisting of *usage scores* and *health scores* (presented at the top right corner in Figure 2). The *usage score* is computed based on the usage of the system by the users, such as logging into the system, recording lifestyle, setting goals, online reading, and forum participation. Meanwhile, the health score is calculated based on the goals achieved and the amount of physical activity carried out by the users. The health score would increase if the blood glucose reading improves and the user has completed any selected physical activity for at least 30 min.

The system features graphical charts to summarize users’ health progress, i.e., blood glucose level, physical activity, food, weight, and stress. There is also a multi-axis graph to illustrate the tracked health parameters and compare these values. The users are free to set their personal health goals and strive to achieve them at their own pace. For instance, they can choose any physical activity (e.g., brisk walking, jogging, and gardening) and then set a duration (minimum 30 min) based on what they aim to achieve. Figure 3 shows the performance bar demonstrating the progress of physical activity carried out by a user.

The system also allows users to set their own goals and plan their tasks accordingly. For blood glucose tracking, they can choose from any of the following: fasting, 2-h post-meal, oral glucose tolerance test (OGTT), and hemoglobin A1c (HbA1c) testing. The line graph, as shown in Figure 4, presents the actual blood glucose level and desired blood glucose level to educate users on how far they are from their target level. A graphical summary of blood glucose readings is also available to conveniently comprehend their progress. Users are able to select the view of the readings; it could be either weekly or monthly, or they could input a specific range of dates.

## 3. Evaluation

### 3.1. Methodology

i-PreventDiabetes was evaluated using a multi-method approach involving 50 participants with prediabetes aged between 19 and 75, between November 2017 and January 2018. The participants were recruited through advertisements shared on social media and posted on notice boards of selected healthcare institutions. They were selected randomly from the public, as they volunteered to participate in this study. Table 4 illustrates a summary of the demographic profile of the study participants. In total, eight out of 50 participants reported using a self-care tool but did not specifically state what type of tool.

Participants were encouraged to use i-PreventDiabetes (which was made available over the Web) at their own pace for six weeks, depending on their requirements and availability. The primary tasks in the system (blood glucose, food, weight, stress, and physical activities) were logged in the system for analysis purpose. Participants were asked to complete questionnaires at the beginning of the study and at the end of the sixth week of the study. Table 5 summarizes the scales used to evaluate i-PreventDiabetes, their purpose, and when they were administered in the study. The statistical package for the social sciences (SPSS) tool (version 25) was used for statistical analysis of the results obtained. At the end of the quantitative study, a short interview was conducted with a purposive sample of eight participants to obtain further insights into their experience with and views of i-PreventDiabetes. According to Six and Macefield, five to 10 participants are sufficient to determine most of the usability problems of a system [30]. The eight interviewees comprised of four participants who used the system extensively and four others who made little use of the system.

Both the TTM and MHLC were administered twice, that is, at the beginning and at the end of the study. The TTM consisted of 24 statements with five response choices, ranging from strongly disagree (1) to strongly agree (5). Participants were expected to respond to the TTM statements to determine if there were any differences in their stage of behavior change toward T2D prevention before and after using the system for six weeks. At the end of the study, the TTM was administered again to see if there was any difference in their stage of behavior change toward diabetes prevention. The McNemar–Bowker test and the Wilcoxon signed-rank test were used to compare the participants’ initial stage and final stage. The statistical significance was set at *p* < 0.05. The MHLC consists of three subscales: internal health locus of control (IHLC), powerful others health locus of control (PHLC), and chance health locus of control (CHLC). There were 18 statements with six answer options ranging from strongly disagree (1) to strongly agree (6). The MHLC was administered pre- and post-study to note any difference in participants’ perception whether health is controlled by internal or external factors. The change scores for each MHLC subscale were then computed by subtracting the baseline and follow-up values.

The final questionnaire of the evaluation study comprised of three scales (IMI, TAM, and UEQ), and this was used only once after the participants had used i-PreventDiabetes for six weeks. The IMI was adopted from the self-determination theory [34]. This scale comprised of 20 items rated on a seven-point Likert scale, from completely untrue (1) to completely true (7). It was employed to identify the motivation level of users in using the system based on five (out of seven) relevant factors: interest/enjoyment, perceived competence, effort/importance, pressure/tension, and value/usefulness. Using the TAM, the perceived usefulness (comprising of 10 statements) and perceived ease of use (comprising of nine statements) of i-PreventDiabetes were assessed. Each TAM question was accompanied by a seven-point Likert scale, from one (strongly disagree) to seven (strongly agree). The UEQ scale comprised of 26 items to measure user experience of the system from six different perspectives, including attractiveness, perspicuity, efficiency, dependability, stimulation, and novelty of the system. Each item of the UEQ was assessed using a five-point Likert scale, ranging from very difficult (1) to very easy (5).

### 3.2. Results

The majority of the study participants were female (70%). Throughout the six weeks, 1500 logins were recorded with a mean (µ) = 37.5. The total number of tasks performed by all the participants is 3639. Figure 5 shows the details of the recorded activities. Food and physical activities were the most frequently recorded tasks, indicating that users primarily use the system to track their food intake and physical activities. At the end of the study, 40 participants completed the six-week evaluation, which had a success rate of 80% participation (40 out of 50 participants). The remaining 10 participants did not participate until the end of the six weeks due to their other responsibilities and work engagements.

The six-week study shows that there were changes in behavior among the participants. Table 6 shows the descriptive values of the initial (IniStage) and final (FStage) behavior change stages. As per our findings, the mean (µ = 3.93) of FStage is higher than the mean (µ = 3.200) of IniStage. This indicates that the participants’ behavior changed at the end of the study, as the responses shifted toward the maximum value on the Likert scale, with higher 50th and 75th percentile values. The *p*-values of the Wilcoxon signed-rank test (<0.001) also indicate that the change in the behavioral change stages of the participants is statistically significant.

Moreover, this study also reveals that the changes in the behavior stages were more positive than negative. Table 7 depicts the results of the ranks of the stages. In total, 17 participants were found to have advanced in the behavior change stages, whereas four participants regressed. The other 19 participants neither progressed nor regressed, they maintained their behavior stages. This contributes to the conclusion that i-PreventDiabetes has the potential to change people’s behavior toward better health management.

Based on the results, some changes were noted in the participants’ attitudes toward the end of the study. Table 8 shows the difference between participants’ responses before and at the end of week six for that subscale. The participants became more aware that their health control comes from within themselves (IHLC), and not from others (PHLC), or their health conditions do not happen by chance (CHLC). The final IHLC mean (µ = 5.021) is higher than the initial IHLC mean (µ = 5.004), whereby the mean values of initial PHLC (µ = 3.346) and CHLC (µ = 2.725) were higher than their final values (PHLC µ = 3.262 and CHLC µ = 2.604). This shows that there were changes in the attitude of the participants toward their healthcare. This could indicate that participants have instilled the self-care habit in them to a certain level.

The participants’ level of motivation to use i-PreventDiabetes indicates that they were pleased with the system. Table 9 shows the mean values of the selected five IMI subscales. The participants felt that the system has value, it is useful for their healthcare management (µ = 5.338), and their level of interest and enjoyment in the system is also quite high. In addition, the results show that the participants imposed less pressure and tension during their involvement with the system. Based on the results, it can be interpreted that the participants were motivated to use the system.

Based on the results, it was revealed that the participants found the system to be useful and user-friendly. Table 10 depicts the results of the TAM scale. Both the subscales of TAM have reasonable mean values to indicate the perceived usefulness (µ = 5.258) and ease of use (µ = 5.345) of the system. The participants agree that using i-PreventDiabetes improves the quality of their lives. It also improves their health performance and effectiveness in caring for their health. Besides making their health easier to manage, it also gives them more control over their lives. Moreover, the participants felt that it was easy to get the system to do what they wanted to do and to remember how to operate the system.

As per the UEQ results, it was found that the participants experienced the system in a positive manner. Figure 6 illustrates the UEQ results and benchmark, which compares i-PreventDiabetes with the other 246 products [38]. Benchmarking analysis of UEQ compares the calculated mean score of i-PreventDiabetes with the result of other items such as websites, online retailers, social media networks, and business applications on the market that were evaluated using this scale. By comparing the evaluated product’s results to the benchmark’s data, conclusions can be drawn about the evaluated product’s relative quality. Based on the benchmark, dependability, stimulation, and novelty are rated as “excellent,” while the other three scales hold “good” ratings.

All the six subscales of UEQ were above 0.8 and have positive values. According to UEQ, values above 0.8 are considered positive, −0.8 to 0.8 neutral, and below −0.8 negative values [37]. The lowest value is 1.431 (novelty), while the highest value is 1.750 (attractiveness and stimulation). The highest value of 1.750 shows that the participants liked the system. It also seems that it is easier to get familiarized with the system. Nevertheless, a few participants felt that the system could be much more creative and innovative.

### 3.3. Discussion

Based on the results, it can be concluded that i-PreventDiabetes is well accepted and supported by the target users. The analysis shows that the system has great potential to improve the participants’ behavior toward T2D prevention. The majority of the study participants (40 of the initial 50) had continuously used the system for six weeks and were keen on the idea of having one system that comprised all the necessary features to prevent or delay diabetes. They also preferred setting their own health goals and meeting them by using selected features of the system at their own pace. Moreover, 32 out of 40 active participants (80%) mentioned that they would like to continue using i-PreventDiabetes after the study. Although the majority of research participants were female, women are less likely to adhere to a healthy lifestyle, despite their increased awareness of the benefits of healthy living [39]. Women’s lack of adherence to healthy lifestyle is mostly associated with socioeconomic factors, as healthy lifestyles are expensive and time-consuming, and women mostly spend their time fulfilling their social roles in the family [40]. This system also positively affected the attitude of people with prediabetes in terms of managing their own health. Their attitude has shifted to the point where, after six weeks, they were more convinced that their health control comes from within rather than from others or by chance. This indicates that systems, such as i-PreventDiabetes, can instill self-care habits among users.

The study participants were also motivated to use the system. They enjoyed using the system and also acknowledged the value and importance of the system. Generally, the participants felt less tense when using the system. In addition, i-PreventDiabetes has been found to be useful in managing the participants’ health. Most of the participants found the system easy to use, although a few of them thought that the system could be further simplified. Goal setting and activity planning have been determined to play a vital role in self-care applications. Evidence indicates that such a feature would foster users to take responsibility for their health [41]. The participants prefer to set their own “achievable” goals and meet the targets by executing the tailored activities using the application. Achievable goals can also motivate the participants; this would make them commit to their goals as they are not being pressured due to strict goals.

Moreover, the user experience of the system is noted to be positive. A few participants shared that the system acts as a constant reminder that triggers them to monitor their blood glucose levels actively. The participants found the system to be convenient. Overall, the study shows that i-PreventDiabetes fosters positive changes in users to prevent diabetes. It is a well-designed system that incorporates the perspectives of both people with prediabetes and healthcare professionals; further, it can enable users to prevent or delay diabetes, provided that the user takes responsibility for making effective use of the features offered in the system. The inclusion of healthcare professionals is suggested to foster people with prediabetes to advance rapidly through behavioral stages in the process of lifestyle modification [42]. i-PreventDiabetes, being a self-care system, did not include healthcare professionals directly in the process of using the system to prevent the progression of prediabetes to diabetes. Results of the evaluation study reveal that it is possible to develop a system that is acceptable by target users provided the design incorporates perspectives of both the target users and healthcare professionals. However, some of the participants expressed that the system should be more autonomous and attractive. According to them, the system should be configured in order to accept and store automatic inputs of their physical activity details from external devices. They also wanted more automatic alerts and reminders from the system.

Furthermore, it was found that participants were a bit skeptical about using the social support features in the system as they did not feel good about sharing their information with others. The users’ attitudes toward receiving social support to help them achieve their health goals were mixed. Social support is often seen as an essential component of health management systems, although it may raise privacy issues. It has been observed that users were not comfortable with communicating with other users or strangers on the system. A cultural difference may be a factor. Users in the United States, for example, have welcomed platforms, such as PatientsLikeMe, where users are eager to freely disclose their symptoms and treatments as well as participate in conversations with other patients dealing with similar health problems [43].

Diabetes prevention systems, such as i-PreventDiabetes, have immense potential and a role in preventing the progression of prediabetes to diabetes. Potential users of such systems are more likely to leverage these systems for their healthcare if they are endorsed and promoted by diabetes institutes, healthcare institutions, hospitals, and wellness centers [44]. Such interventions can complement or be integrated with existing T2D prevention programs.

## 4. Conclusions

Prevention of T2D is possible with proper diet and lifestyle modification at the prediabetes stage. Digital technology enables health consumers to prevent or delay diabetes by engaging them in self-care tasks through interventions that foster behavior change toward better health. In this study, we proposed a novel framework for self-care systems developed for prediabetes and designed a prototype called i-PreventDiabetes targeted at health consumers, which was well accepted by people with prediabetes.

Our study has shown that individuals with prediabetes can be empowered to lower their blood glucose levels by providing them with interventions that allow them to alter their lifestyles. The results demonstrate that the i-PreventDiabetes framework and application is feasible to promote behavior change among individuals with prediabetes. Through this application, users realize that their health control comes from within themselves and not from others, nor happens by chance. The application was also found to be useful and easy to use. Users are motivated to use the application and experienced it positively. Overall, the results suggest that i-PreventDiabetes has the potential to instill the right behavior among people with prediabetes to modify their lifestyle and diet to prevent T2D. The proposed framework could serve as a guide for system developers and researchers to develop practical self-care applications for T2D prevention.

### 4.1. Future Work

The further success of this application depends on the larger user community and developer support. The application prototype demonstrated the integration of behavioral change theories to promote the prevention of T2D and confirmed that the proposed framework is comprehensive, meets the needs and expectations of the target users, and aids in instilling the right attitude in users toward preventing T2D. Longer-term studies with a larger number of users and additional lifestyle changing features are necessary to validate and quantify the long-term health effects of i-PreventDiabetes. Further studies can be carried out to leverage the potential of the internet of things (IoT) for autonomous capture of vital signs of users. Big data can be applied to enhance the intelligence of the system in predicting user behavior toward health, and features included can be tailored to achieve desired health goals.

### 4.2. Limitations

The vision of i-PreventDiabetes covers a large scope; thus, only essential features of the framework were implemented and tested. A long duration is required to determine more effective results for behavior change. It takes six months for a person to change from one stage of behavior to another [25]. The effectiveness of the application prototype in reducing the blood glucose level has not been evaluated because it is too short a time to evaluate its effectiveness.

Most participants of the evaluation study had experience with computers, and results for users unfamiliar with computers may differ. We did not provide participants with the necessary equipment, such as glucometers, to track their vital sign data. Furthermore, we administered the MHLC scale at different stages of the study to compare changes in their measures, but there was no control group for comparison.

## Figures and Tables

**Figure 1 ijerph-18-09160-f001:**
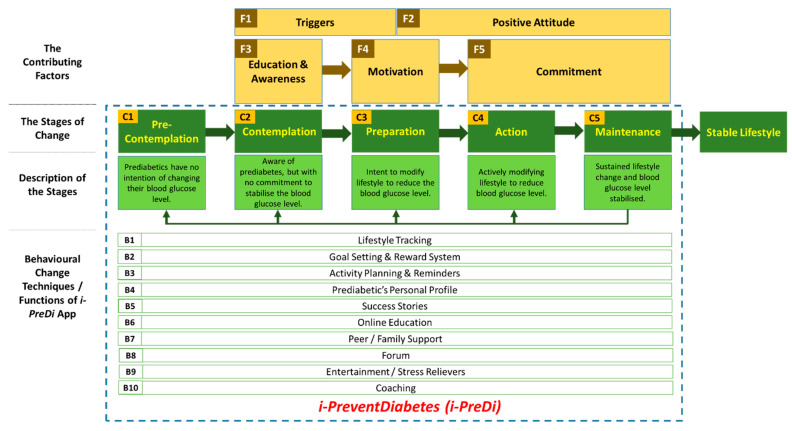
Conceptual framework for prediabetes self-care systems.

**Figure 2 ijerph-18-09160-f002:**
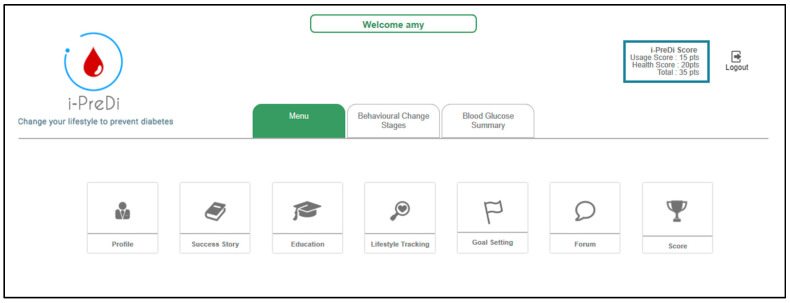
The dashboard of i-PreventDiabetes.

**Figure 3 ijerph-18-09160-f003:**
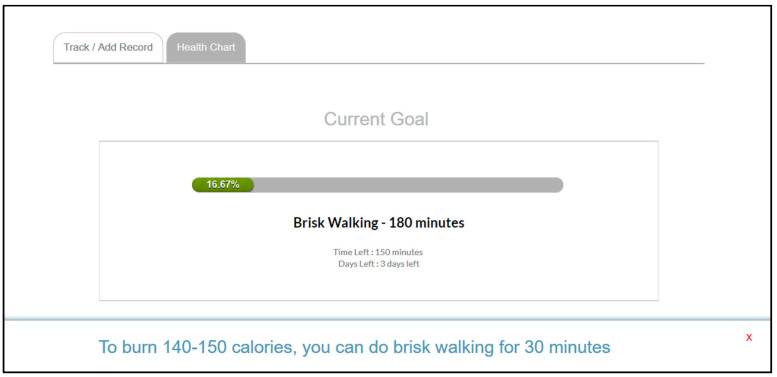
A performance bar showing the progress of the physical activity completed by a user.

**Figure 4 ijerph-18-09160-f004:**
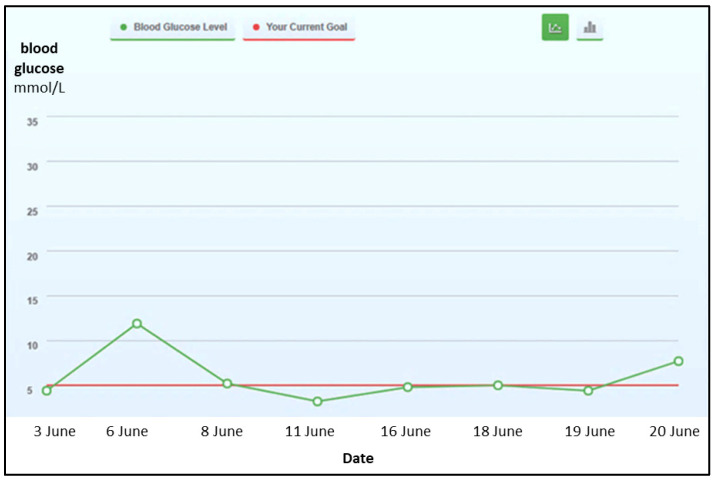
A line graph showing the actual blood glucose level and the desired blood glucose level (fasting blood glucose).

**Figure 5 ijerph-18-09160-f005:**
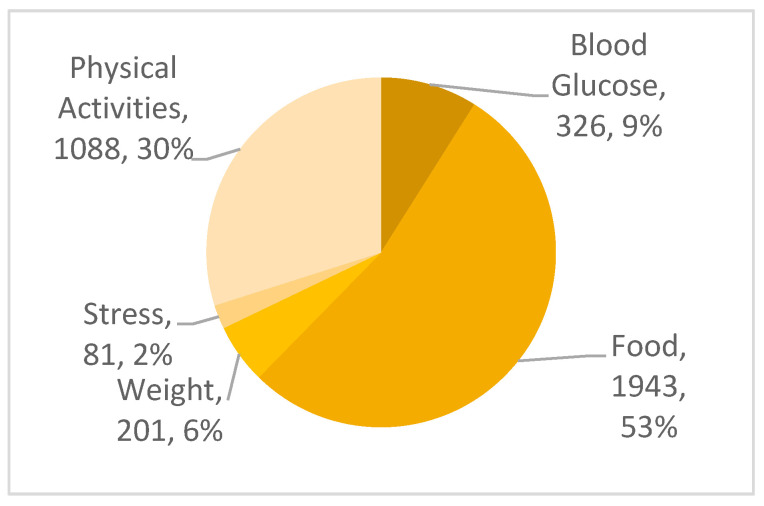
The frequency of tasks recorded in i-PreventDiabetes.

**Figure 6 ijerph-18-09160-f006:**
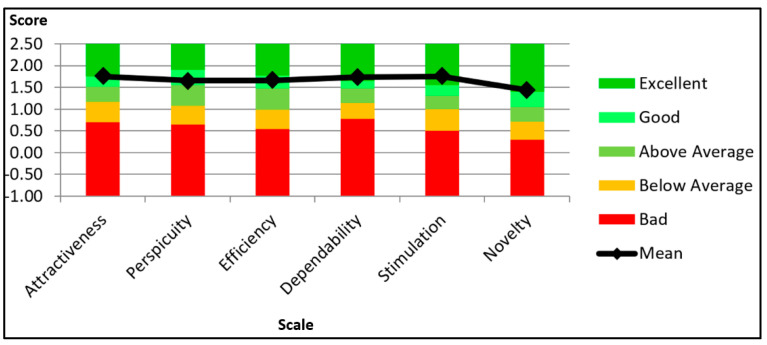
The UEQ results and benchmark.

**Table 1 ijerph-18-09160-t001:** The stages of change.

Code	Stages	Description
C1	Pre-Contemplation	Non-believers: Do not believe in changing behavior. Not ready and no intention of changing behavior.
		Believers: Believe in changing behavior. But not ready and with no intention of changing behavior.
C2	Contemplation	Getting ready to change behavior. The prediabetic is aware of his/her prediabetes but with no commitment to stabilize the blood glucose level.
C3	Preparation	Ready to change behavior. The prediabetic is intent to modify his/her lifestyle to reduce the blood glucose level.
C4	Action	The prediabetic has made some specific lifestyle changes to reverse the progression to diabetes.
C5	Maintenance	The prediabetic has made some specific lifestyle changes and is working to prevent relapse.

**Table 2 ijerph-18-09160-t002:** The contributing factors.

Code	Factors	Description
F1	Triggers	Something that causes an individual to change their lifestyle by raising awareness or creating a need to change. There could be life-changing incidents/events which happen to them, their family members, or their friends. It could also be something they read or heard somewhere (e.g., signboards, brochures, TV, social media) regarding diabetes.
F2	Positive Attitude	The prediabetic’s attitude toward lifestyle (e.g., “make it happen” attitude) and the determination to change their lifestyle to reduce the blood glucose level.
F3	Education & Awareness	Education and awareness of prediabetes and diabetes and the influence of lifestyle in the life of a prediabetic generate the desire for change.
F4	Motivation	The prediabetic’s eagerness and readiness to alter his or her lifestyle.
F5	Commitment	The prediabetic is dedicated to taking the necessary actions to improve lifestyle and then maintain a healthy lifestyle through continuous engagement.

**Table 3 ijerph-18-09160-t003:** BCTs presented as features in i-PreventDiabetes.

Code	Functions	Description
B1	Lifestyle Tracking	To track blood glucose readings, physical activities, count calories of food consumed, the amount of carbohydrates consumed, food intake, weight, and stress level. Readings will be presented using visuals to show how far they are from the desired target level.
B2	Goal Setting & Reward System	To set desired and achievable lifestyle goal(s) based on the items measured in B1 for a particular duration (e.g., 3 months, 6 months, 1 year). If succeeded, score points will be given.
B3	Activity Planning & Reminders	Planning of tasks (e.g., 20 min of brisk walking every day, 10,000 steps in a day, cut down carb intake) to achieve the desired goals. Reminders will be sent for each of the task planned.
B4	Prediabetic’s Personal Profile	Prediabetic to have his/her own profile with information, such as name, age, gender. Current health status of the prediabetic, the stages of change, and the score points will be updated here.
B5	Success Stories	Display of success stories of other individuals with prediabetes, where they manage to stabilize their blood glucose level because of the lifestyle changes they have made.
B6	Online Education	Education about prediabetes, lifestyle changes, calorie content in each type of food, type of physical activities to burn calories, etc.
B7	Peer/Family Support	Support by family or friends, in the form of text messages or e-mail, as reminders and as a companion to do physical activities and managing the type of food they eat.
B8	Forum	A platform where people with prediabetes can communicate with each other to share what worked and what did not work and support each other in their lifestyle changes.
B9	Entertainment/Stress Relievers	Sharing of educational materials to reduce stress level (e.g., jokes, cartoons, YouTube videos, games).
B10	Coaching	Individuals with prediabetes can communicate with their healthcare professionals to discuss their health.

**Table 4 ijerph-18-09160-t004:** Demographic characteristics of the study participants.

Characteristic	*N*	%	Characteristics	*N*	%
Age (years)			Family history of diabetes		
Below 20	1	2	Yes	32	64
21–30	9	18	No	18	36
31–40	9	18	Computer usage		
41–50	10	20	5+ days/week	34	68
51–60	15	30	1–4 days/week	10	20
61 and above	6	12	1–5 times/month	4	8
Gender			Few times/year	1	2
Male	15	30	Never	1	2
Female	35	70	Uses a self-care tool (Samsung/iPhone Health app, health tracker, step counter, etc.)		
Ethnicity			Yes	8	16
Malay	10	20	No	42	84
Indian	34	68			
Chinese	5	10			
Others	1	2			

**Table 5 ijerph-18-09160-t005:** Scales administered to evaluate i-PreventDiabetes.

Scales	Purpose	
Transtheoretical model (TTM) [24]	To study the influence of i-PreventDiabetes in fostering behavior change among the users.	Pre- and post-study
Multidimensional health locus of control (MHLC) [31,32]	To investigate whether i-PreventDiabetes can positively affect the users’ attitude toward managing their health.	
Intrinsic motivation inventory (IMI) [33,34]	To evaluate users’ subjective experience (levels of intrinsic motivation) in their interaction with i-PreventDiabetes.	Post-study
Technology acceptance model (TAM) [35,36]	To determine the acceptability of i-PreventDiabetes.	
User experience question (UEQ) [37]	To measure the user experience of i-PreventDiabetes.	

**Table 6 ijerph-18-09160-t006:** Descriptive values of the initial and final behavior change stages of the TTM scale.

	*N*	µ	σ	Minimum	Maximum	Percentiles		
						25th	50th (Median)	75th
IniStage	40	3.20	1.265	1	6	3.00	3.00	3.00
FStage	40	3.93	1.509	1	6	3.00	4.50	5.00

**Table 7 ijerph-18-09160-t007:** Results of the ranks for the behavior change stages of the TTM scale.

		*N*	Mean Rank	Sum of Ranks
FStage—IniStage	Negative ranks	4 ^a^	4.00	16.00
	Positive ranks	17 ^b^	12.65	215.00
	Ties	19 ^c^		
	Total	40		

^a^ FStage < IniStage. ^b^ FStage > IniStage. ^c^ FStage = IniStage.

**Table 8 ijerph-18-09160-t008:** Changes in user response to the MHLC subscales within the six-week period between user surveys (*n* = 40).

Subscale		µ	σ	*p*-Values
Pair 1	IHLC_Initial	5.004	0.574	0.017
	IHLC_Final	5.021	0.566	
Pair 2	PHLC_Initial	3.346	1.070	0.000
	PHLC_Final	3.262	1.014	
Pair 3	CHLC_Initial	2.725	0.804	0.001
	CHLC_Final	2.604	0.769	

**Table 9 ijerph-18-09160-t009:** Mean (µ) and standard deviation (σ) value of selected IMI subscales.

Scheme	*n*	µ	σ
Interest/enjoyment	40	5.287500	1.2526254
Perceived competence	40	4.993750	1.2589265
Effort/importance	40	4.400000	1.0250078
Pressure/tension	40	2.675000	0.8208032
Value/usefulness	40	5.337500	1.4692903

**Table 10 ijerph-18-09160-t010:** Mean (µ) and standard deviation (σ) value of TAM.

Subscale	*n*	µ	σ
Perceived usefulness	40	5.258333	1.3111612
Perceived ease of use	40	5.345000	1.0307951

## Data Availability

The data presented in this study are available on request from the corresponding author. The data are not publicly available to maintain the privacy and confidentiality of the participants involved.

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
