# Peer review of "Behavioral Theory-Based Framework for Prediabetes Self-Care System—Design Perspectives and Validation Results"

_ijerph, 2021, doi:10.3390/ijerph18179160_

Round 1
Reviewer 1 Report
This is a nice manuscript that propose a framework to aid the development of self-care systems to prevent diabetes, which integrates behavioural change theories and techniques and offers features such as goal setting, activity planning, and health monitoring.
The manuscript is very interesting and deals with a very important topic: however I have some comments:
The first concern is related to patient’s selection. In this preliminary report the number of females is greater than the number of men (70%versus 30%) (See table 4)
Due to the fact that there is a gender bias related to diabetes as well as cardiovascular disease, it would be appropriate to include a comment on this high number of women. Useful to quote
" Cocchi C, Coppi F,et al. Cardiovascular disease prevention and therapy in women with Type 2 diabetes. Future Cardiol. 2021 Mar 19. doi: 10.2217/fca-2021-0011. Epub ahead of print. PMID: 33739145”
“Humphries KH, Izadnegahdar M, Sedlak T et al. Sex differences in cardiovascular disease – impact on care and outcomes. Front. Neuroendocrinol. 46, 46–70 (2017).”
Specifically, women are more aware of the positive effects of a healthy lifestyle but nevertheless are less adherent to the healthy lifestyle. The reasons are mostly socio-economic, because healthy lifestyles are expensive and time-consuming and women have a lot of time occupied by social roles in the family. You may find this reference useful: “Mattioli AV, Sciomer S, Maffei S, Gallina S. Lifestyle and Stress Management in Women During COVID-19 Pandemic: Impact on Cardiovascular Risk Burden. Am J Lifestyle Med 2021, 15(3), pp. 356–359
doi: 10.1177/1559827620981014”
Finally some practical suggestions on how to encourage patients to use this algorithm would improve the manuscript. Do you think it should be suggested by the prevention center? or from social advertising campaigns?
Tables and figure are very good
Author Response
Thank you very much for your feedback and comments provided to improve the manuscript. We have amended the manuscript accordingly. Please find attached a document comprising of the correction tables (addressing all reviewers' comments) and evidence that the work was proofread.

Reviewer 2 Report
This is a good paper about preventing diabetes and it is very important for public health; however, the following should be considered to improve the quality of the paper.
- Page 6: Measuring different types of physical activity should be explained in detail. It is not clear if the participant does other activities rather than brisk walking how it would be scored. Were the participants restricted to just do brisk walking? If so it needs to be stated.
- Figure 4: It is not clease which blood glucose was measured, is it fasting blood glucose? What is the unit of measurement for blood glucose? The figure needs vertical and horizontal captions. Does this show blood glucose during a day, week, or month?
- Table 4 needs clarification, what self-care tools were used by participants? Were they related to diabetes?
- It is not clear why only 8 participants were selected for a short interview, what were the charactristics of chosen participants?
- Abbreviations should be defined before use in the text, please define all the abreviations
- All the questionnaires and surveys should be refrenced or attached to the paper.
- More details are needed regarding the statistical software/s that was/were used for data analysis. Name, version,....
- What is the reported mean under the results section? Is this per participant?
- I suggest substitute activities with another word, it confuses readers as if this is for physical activity or all the variables
- Page 9: "This indicates that the participants’ behavior changed at
the end of the study, as the responses shifted toward the maximum value on the Likert scale, with higher 50th and 75th percentile values" this is not a correct statement since there is no mid point measurement, participants might have changed their behaviour right after they started participating in the study. Please repherase. - Table 7 a, b, and c need to be defined. This table need more details, what survey/s was/were used for this?
- "1.FStage < IniStage; 2.FStage > IniStage; 3.FStage = IniStage". This is not clear, please explain what does this mean?
- How were the results compared to other 246 products? Need to give more details about the other products and the method for comparison.
- Figure 6 need vertical caption.
- "The more they use the system, change in their stage of behaviour toward diabetes prevention is noted." This statement is not correct since the measurements are just pre and post. More time points are needed for this conclusion.
Author Response

(The authors gave the same response as above.)

Reviewer 3 Report
ABSTRACT:
Rephrase the sentence “Prediabetes is a stage in which blood glucose levels are elevated but not to the point of diabetes.”
Prediabetes CAN BE reversible and dietary modifications are included in lifestyles changes.
No diabetes, is type 2 diabetes.
"prediabetic" should not be used to define people with prediabetes
MANUSCRIPT
Diabetes is not type 2 diabetes.
Ref 1 obsolete. Also 2-4 ref.
Need ref for “Given that many people with prediabetes do not take their conditions seriously or lack the self-motivation”
No blood sugar levels, glycaemia or blood glucose levels
If i-PreventDiabetes is based on acceptance stages it’s necessary to introduce it on introduction section. Are there acceptance / action phases? What are they based on? What articles support it?
It is necessary to define well the acronyms used (BCT, ADEE7, etc), also the words that define a concept should be the same (sometimes sugar levels, sometimes glucose levels)
The discussion is a summary of the results. It is necessary to reformulate it indicating the importance of the results obtained based on other published studies. For example, what other studies are there on applications (in prediabetes or type 2 diabetes) and if they obtained similar results, what do these data suggest if they were used in the prevention of T2D, etc.
Author Response

(The authors gave the same response as above.)

Reviewer 4 Report
The obsolescence rate of the bibliographic references is 7 years (high for this type of articles). Only 41.86% of the references are 5 years old or less. Therefore, we recommend updating the bibliographic references.
Reference 42 should be adapted to the standards of the journal.
Author Response

(The authors gave the same response as above.)

Round 2
Reviewer 2 Report
The quality of the manuscript has been improved significantly.
Reviewer 3 Report
Thank your for all the modifications. The manuscript has improved remarkably.